# Hypoxia favors tumor growth in colorectal cancer in an integrin αDβ1/hemoglobin δ-dependent manner

Erkki Koivunen[1],[*] , Sudarrshan Madhavan[1] , Laura Bermudez Garrido[1] , Mikaela Grönholm[2] , Tuomas Kaprio[3,4] , Caj Haglund[3,4], Leif C Andersson[5] , Carl G Gahmberg[1],[*]

**Low oxygen tension (PO$_2$), characterizes the tissue environment of tumors. The colorectal tumor line Colo205, grown under reduced oxygen tension expresses a novel αDβ1 integrin, which forms a cell surface complex with hemoglobin δ. This resulted in high local affinity for oxygen, which increased cell adhesion as compared with cells grown under normal oxygen tension. Staining with antibodies to the integrin αD polypeptide and hemoglobin δ, and transfection with cDNAs for GFP-hemoglobin δ and mCherry-αD, showed co-localization of αD and hemoglobin δ. Antibodies to αD and β1 integrins, an RGD peptide, and an αDβ1 binding peptide from hemoglobin δ, blocked the αDβ1-hemoglobin interaction and lowered oxygen consumption. Downregulation of integrin αD or hemoglobin δ expression inhibited cell proliferation in hypoxia. The very frequent expression of complexes between αDβ1 and hemoglobin δ on the cell surface offers potential diagnostic and therapeutic targets in colorectal cancer.**

## Introduction

Neoplasms arise in an environment of normal cells that interact with the surrounding cells and the extracellular matrix. At early stages, tumors that require oxygen and nutrients for their proliferation, exist under low oxygen tension with a high interstitial fluid pressure (Ferretti et al, 2009; Muz et al, 2015; Donnem et al, 2018).

Relatively little is known about how cell adhesion and oxygen uptake is controlled in malignant cells. Low oxygen may induce the synthesis of transcription factors such as hypoxia-induced factors (HIFs) or Kruppel-like factors (KLF), but their possible effects on cancer cells have remained limited.

To examine the expression of relevant proteins under hypoxia, we analyzed the mRNAs expressed in colorectal carcinoma cells, focusing on integrins and Hbs. Interestingly, the cells expressed mRNAs for the integrin αD polypeptide, Hbδ and Hbβ, but surprisingly not for the integrin β2 chain, known to form a functional integrin with the αD chain.

Integrins are protein heterodimers formed by α- and β-chains. Most cells express integrins and they form a major family of adhesion proteins (Hynes, 2002; Bachmann et al, 2019; Kechagia et al, 2019; Gahmberg & Grönholm, 2022). Cancer cells mostly express integrins that belong to the β1 integrin subfamily (Ruoslahti & Giancotti, 1989; Desgrosellier & Cheresh, 2010; Ju et al, 2017). Integrins may bind to both cellular and extracellular matrix ligands. Many integrins bind to the arginine-glycine-aspartic acid (RGD) motif present in fibronectin, fibrinogen, and several other proteins (Pierschbacher & Ruoslahti, 1984). Some integrins, including the three leukocyte β2 integrins LFA-1 (CD11a/CD18, αLβ2), Mac-1 (CD11b/CD18, αMβ2) and p150,95 (C11c/CD18, αXβ2) bind to large protein domains in their intercellular adhesion molecule ligands (ICAMs) (Staunton et al, 1988; Li et al, 1993). The fourth β2 integrin, αDβ2 (CD11d/CD18) was more recently discovered, and it has been less studied (Van der Vieren et al, 1995). Monocyte/macrophage cell lineages express αDβ2 (Podolnikova et al, 2016; Aziz et al, 2017; Blythe et al, 2021) and a known cellular ligand is ICAM-3 (Van der Vieren et al, 1995).

Humans express several hemoglobins (Hb) during development (Sankaran et al, 2010), which include embryonic, fetal and adult Hbs, which have different capabilities to form dimers and tetramers (Manning et al, 2007). Embryonic and fetal Hbs have a higher affinity for oxygen than adult Hb$α_2β_2$ (HbA1). Hbζ and Hbε are down-regulated in early embryos, and later replaced by tetrameric fetal Hb$α_2Υ_2$. Hb$α_2Υ_2$ is then replaced by Hb$α_2δ_2$ (HbA2), which is expressed in fetal liver megakaryocyte-erythrocyte restricted progenitors, and it is later down-regulated to 2–3% of total Hb (Sankaran et al, 2010).

Malignant cells express Hbs. For a review, see Pedro (2023). K562 leukemia cells have been extensively studied, and shown to express embryonic and fetal Hbs, including Hbδ, but not Hbβ

[1]Programme in Molecular and Integrative Biosciences, Faculty of Biological and Environmental Sciences, University of Helsinki, Helsinki, Finland   [2]Drug Research Program, Division of Pharmaceutical Biosciences, Faculty of Pharmacy, University of Helsinki, Helsinki, Finland   [3]Programme in Translational Cancer Medicine, Faculty of Medicine, University of Helsinki, Helsinki, Finland   [4]Department of Surgery, University of Helsinki and Helsinki University Hospital, Helsinki, Finland   [5]Department of Pathology. Faculty of Medicine, University of Helsinki, Helsinki, Finland

Correspondence: carl.gahmberg@helsinki.fi
*Erkki Koivunen and Carl G Gahmberg contributed equally to this work

(Andersson et al, 1979; Rutherford et al, 1979; Dean et al, 1983; Poddie et al, 2003). Hb has been shown to be expressed in thyroid cancer (Onda et al, 2005), breast cancer (Ponzetti et al, 2017), renal cancer (Kurota et al, 2023), and glioblastoma (Emara et al, 2014), but possible functions have not been studied.

When studying hypoxia effects on cellular localization and function of integrin αD, and Hb, we found that the αD chain forms a novel integrin αDβ1, which makes a cell surface complex with Hbδ, and this complex is required for colorectal cancer cell proliferation.

Immunochemistry of tumor sections from human colorectal carcinoma patients showed that a high proportion of the tumors stained positively both for the αD integrin and Hbδ, and the proteins showed excellent co-distribution. The adjacent normal tissue was largely negative.

# Results

### Colorectal Colo205 carcinoma cells grown under hypoxia express the integrin αD, Hbδ, and Hbβ

We used the colon cancer cell line Colo205 as a model for colorectal carcinoma. The Colo205 cells were grown under normoxic (20% $O_2$) or hypoxic conditions ($O_2$ gradient 1–5%). After 48 h in hypoxia, the oxygen concentration in the cells was below 5%, as shown using the Image-iT Hypoxia green reagent (Fig S1). Most cells were round when grown under normoxia (Fig 1A), whereas low oxygen tension induced changes in cell morphology, and increased the cell binding to plastic surfaces. The average size of the cells grown under normoxia was 574 arbitrary units, whereas the size of the cells grown under hypoxia was 1767 (Fig 1B). The fact that the cells under hypoxia grew well and bound to underlying surfaces, suggested that cell adhesion and oxygen-binding proteins could be involved.

Qualitative RNA analysis showed that the cells expressed several integrin and Hb mRNAs. Unexpectedly, the cells expressed integrin αD mRNA and Hbα and Hbδ mRNAs, but not integrin β2 mRNA. Hbδ and Hbβ are closely related, differing in only 10 amino acids, and may not be separately identified (Table 1). The αD integrin chain has earlier only been described from leukocytes, where it forms the αDβ2 integrin. The expression of integrin αD and Hbs in the lysates of cells showed an increase in αD in hypoxia-grown cells (Fig 1C). Blots of His-tagged recombinant Hbδ showed under reducing conditions SDS-stable multimers, whereas Hbβ blotted as a monomer (Fig S2A). The visualization of Colo205 derived Hbδ required longer exposure (Fig S2B). A high molecular weight aggregate was visualized (Fig S2B). The hypoxia-grown cells contain high protease activity, and therefore Western blots may underestimate the protein levels. Therefore, we measured the expression of the αD integrin mRNA, and determined which Hbδ/β the cells express. qRT-PCR analyses were performed on cells grown in normoxia and hypoxia. By use of αD, Hbδ, and Hbβ specific primers, and Hb cDNA sequencing, we found expression of both Hbδ and Hbβ. Cells grown under hypoxia showed a 15x increase in αD, and 7-9x increase in Hbδ and Hbβ mRNAs (Fig 1D).

### Cell surface integrin αD chain co-localizes with Hbδ

The immunofluorescence studies of Colo205 cells grown under hypoxic conditions, showed strong αD integrin membrane expression (Fig 2A). β2 integrin staining was negative. Integrin β1 staining was positive, and β1 co-localized with αD integrin (Pearson's correlation coefficient 0.69). Cells grown under normoxia showed weak expression of both αD integrin and Hb (Fig 2B). After 2 h under hypoxia, more αD appeared at the cell surface, but the Hb expression remained low. Both αD and Hb became clearly visible after 9 h, and they showed good co-localization (correlation 0.74). Integrin αD and Hb were expressed after 24 h in cap-like structures. A three-dimensional reconstruction of a cap from a video shows an Hb aggregate surrounded by the αD integrin in the last figure of the expression sequence of Fig 2B (Video 1). Surprisingly, Hbα was not detected at the cell surface. Most Hbs are formed by two α-chains and two β-chains. The result indicated that the cell surface Hb lacks α-chains (see below).

Colo205 cells were then transfected with mCherry-αD integrin and GFP-Hbδ constructs (Fig 2C). The transfection efficiencies were about 50%. The cells showed strong plasma membrane fluorescence for both mCherry-αD and GFP-Hbδ when grown under normoxia, and with high co-localization (0.97). Transfected cells grown under hypoxic conditions expressed αD integrin and Hbδ in membrane caps and Hbδ appeared aggregated, like the case with endogenous proteins. The co-localization was high (0.76). Colo205 cells were also transfected with mCherry-β1 integrin (Fig 2D). mCherry-β1 was uniformly distributed on the membrane, and it showed good co-localization with Hbδ (0.81).

### Colo205 cells express in hypoxia the novel integrin αDβ1

The immunofluorescence studies indicated that the integrin αD chain was associated with integrin β1 chain. To prove that the integrin β2 chain is lacking from Colo205 cells, we did immunoblotting of Colo205 lysates with antibodies to integrins β1 and β2. They showed expression of β1 only (Fig 3A), whereas the monocytic U937 control cells expressed a weakly labeled β1 and a more strongly labeled β2 (Fig 3A). The presence of αD and β1 integrin heterodimers was confirmed by immunoprecipitation of Colo205 cells grown under hypoxia with β1 antibodies, followed by immunoblotting of the immune precipitates with integrin αD antibody (Fig 3B). When tested for the presence of a complex of Hbδ and αDβ1 integrin, the Hbδ antibody did not co-precipitate integrin αD from lysates of cells grown under normoxia, whereas αD co-precipitated from lysates of cells grown under hypoxia (Fig 3C).

### Structural analysis identifies binding sites on Hbδ for the αDβ1 integrin I-domain

A schematic figure of the αDβ1 integrin ligand-binding I domain bound to Hbδ is shown in Fig 3D. To elucidate how Hbδ and integrin αD could interact, we used AlphaFold2 to predict the structure of an Hbδ tetramer bound to the αD I-domain (Fig 3E). Hbδ showed two protruding loops covering the Hbδ sequences SFGDLSSP and QLSELHCDKL, capable of interacting with the αD integrin I domain. The Hbδ tetramer model had full confidence (90%), and its

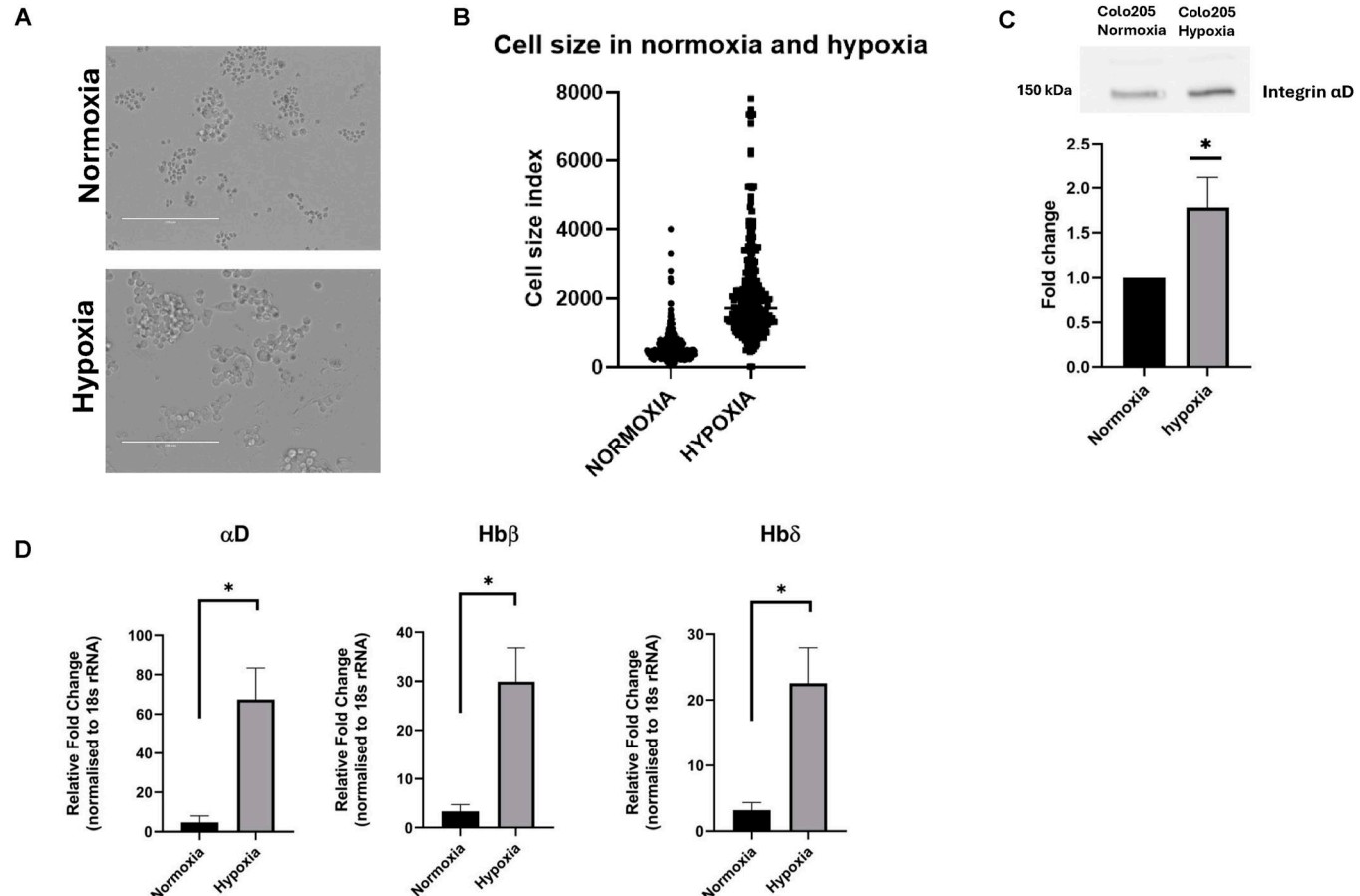

**Figure 1. Morphology of Colo205 cells and expression of integrin αD, Hbβ and Hbδ.**
**(A)** Colo205 cells grown under normal oxygen pressure (above) and for 48 h under hypoxia (below). Scale bar 200 μm. **(B)** Plots of the cell sizes. **(C)** Blots of αD from lysates of Colo205 cells grown under normoxic and hypoxic conditions. *$P$ < 0.05 **(D)** qRT-PCRs of αD, Hbβ and Hbδ from Colo205 cells grown for 48 h under normoxia and hypoxia. *$P$ < 0.05.
Source data are available for this figure.

**Table 1. Integrin and erythroid cell genes expressed in Colo205 cells grown in hypoxia.**

| Integrin α-chain genes | Integrin β-chain genes | Erythroid genes |
|---|---|---|
| α1, α2, α5, α6, α7, α10, αD, αE | β1, β3, β4, β5, β6, β7, β8 | Hbα, Hbδ, ICAM-4 |

Expressed integrin and red cell genes in Colo205 cells grown in hypoxia were studied by RNA sequencing. A qualitative analysis was made as described in MM.

interaction with the integrin I domain good confidence (80%). Computational alanine scanning mutagenesis showed the importance of the aspartate of SFGDLSSP and the glutamine, glutamic acid, and lysine of QLSELHCDKL, as alanine substitutions decreased the confidence of αD integrin binding significantly (aspartate, glutamate) or showed no binding (glutamine, lysine) (data not shown).

Fig 3F and G show the amino acid sequences of Hbδ and Hbβ, respectively. The differences are marked in blue, the locations of the interacting peptide regions of Hbδ in red and green, and that of Hbβ in orange.

## The binding of hypoxia-grown Colo205 cells to Hb is inhibited by antibodies to integrin αD, integrin β1, an RGD peptide, and to fibronectin by soluble Hb

We used Colo205 cell adhesion to plastic plates coated with red blood cell Hb to study the interaction between αDβ1 and surface Hb. Hypoxia-grown Colo205 cells bound well to red blood cell Hb, whereas binding of cells grown in normoxia was low (Fig 4A). Monoclonal antibodies to integrin αD and β1 efficiently blocked the binding of hypoxia-grown cells. The β2 integrin-blocking antibody 7E4 showed no inhibition. The RGD motif-containing fibronectin peptide efficiently blocked adhesion, whereas the control peptide had no effect (Fig 4B). Furthermore proof of the importance of αDβ1 in binding to Hbδ was obtained by transfection of mCherry-αD cDNA into Colo205 cells, which resulted in increased adhesion to Hb (Fig S3). To confirm that Colo205 cells grown under hypoxia bind to Hbδ, we coated plates with recombinant Hbδ. The cells bound well and the β1 antibody and the GRGDSP peptide blocked the binding (Fig S3). We then studied the adhesion to fibronectin and the effects of soluble Hb

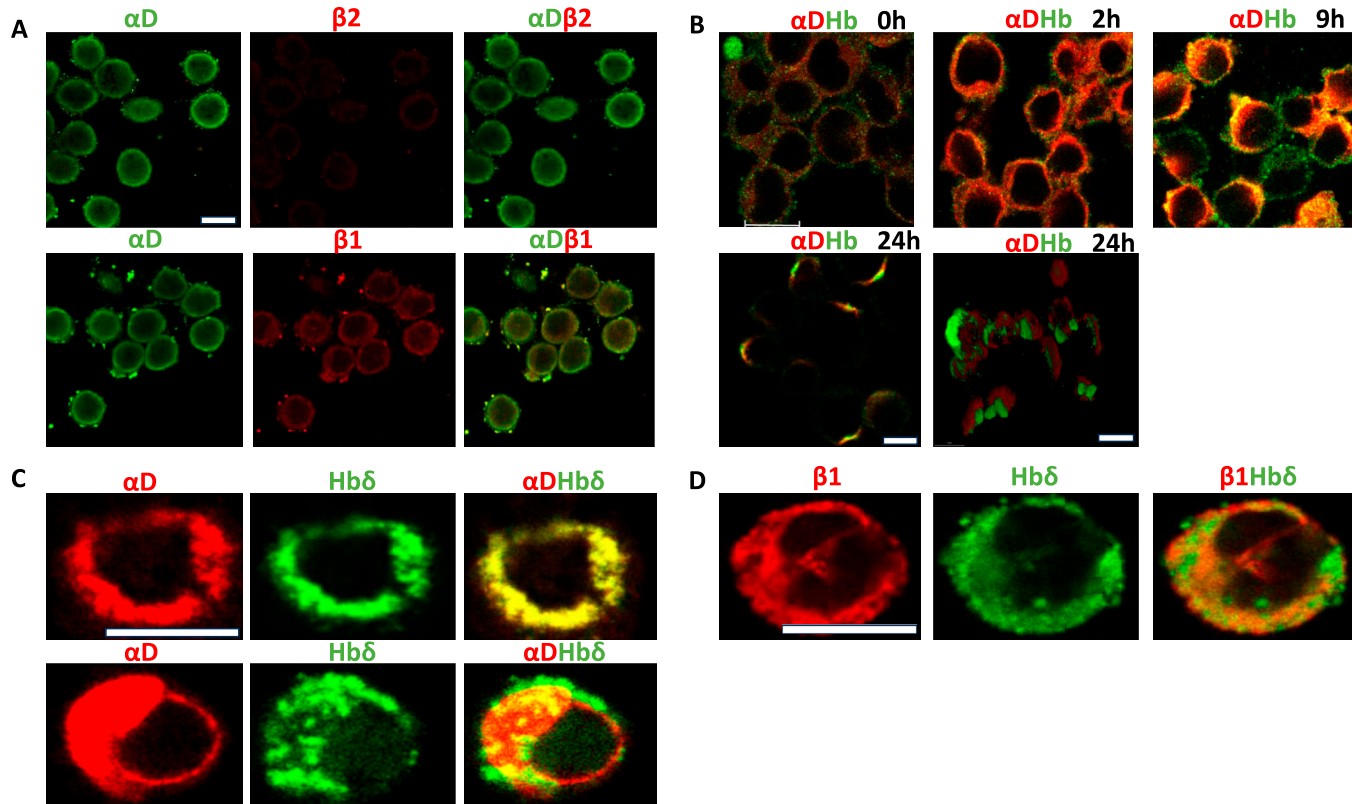

**Figure 2. Immunofluorescence staining of Colo205 cells.**
**(A)** Cells were stained for integrin αD (green, rabbit polyclonal antibody), and β2 integrin (red) chains showed absence of β2. Integrin β1 staining (red) was strong and it co-distributed with αD staining. Scale bar 10 μm. **(B)** Cells were grown in normoxia, transferred to hypoxia, and stained at different time points for αD (red) and Hb (green, goat polyclonal). After 2 h under hypoxia, the cells showed increased staining for αD, but weak staining for Hb. After 9 h, Hb staining was strong and showed partial co-distribution with αD. After 24 h, cells expressed cup-like structures. Scale bar 10 μm. The last figure shows the integrin αD (red) and Hbδ (green) at 24 h under hypoxic conditions. The figure was obtained from the Video 1. Scale bar 10 μm. **(C)** Higher magnification of αD-mCherry and Hbδ-GFP- transfected cells grown under normoxia (above) and hypoxia (below). Scale bar 10 μm. Hbδ formed aggregates and there was more capping in hypoxia-grown cells. **(D)** Cells transfected with integrin β1-mCherry and Hbδ-GFP showed partial co-distribution under normoxia. The magnification is the same as in Fig 2C.

and αD antibody. Both soluble Hb and αD antibody partially blocked the binding of cells grown under hypoxia (Fig 4C).

We also studied to some extent the colorectal cancer cell line HT-29, and it appeared similar to the Colo205 cells. HT-29 cells, grown under normoxia and hypoxia express αD (Fig S4). No quantitation was made.

### Oxygen consumption and cell proliferation are inhibited by Hbδ/β and integrin αD antibodies, the Hbδ derived peptide QLSELHCDKL and by down-regulation of αD and Hbδ

The monoclonal Hbδ/β antibody and the Hbδ-derived QLSELHCDKL peptide, inhibited oxygen consumption by hypoxia-grown Colo295 cells. The corresponding peptide from Hbβ, ATLSELHCDKL showed no effect (Fig 4D). In addition, the oxygen consumption and the proliferation of cells grown under hypoxia, were blocked by anti-bodies against integrin αD, but not by antibodies to the closely related integrin αX (Fig S5).

Hbδ and integrin αD RNAi treatments efficiently blocked Colo205 cell proliferation, whereas Hbβ RNAi treatment had no effect (Fig 4E). The down-regulation of αD was about 70%, whereas the down-

regulation of Hbδ could not be accurately determined due to protein aggregation, when studied by blotting of gels.

### Colorectal tumors co-express integrin αD and Hbδ

Tumor tissue in sections of colorectal cancers stained positively with antibodies to integrin αD and Hbδ. Fig 5A shows strong staining of a tumor with the Hbδ monoclonal antibody, and control staining of the same tumor with mouse IgG was negative (Fig 5B). Fig 5C and D shows staining of another colon cancer tumor with Hbδ (Fig 5C), and Hbα antibodies (Fig 5D). The Hbα antibody stained only red cells in the blood vessels of the normal tissue. Fig 5E and F show two consecutive sections of a colon tumor stained with antibodies to Hbδ and integrin αD, revealing co-distribution of the antigens. The β1 integrin was expressed both in tumor and normal tissues (Fig 5G), but the tumors lacked the β2 integrin (Fig 5H). Analysis of a tissue microarray of colorectal cancers showed an expression frequency of more than 80% for both Hbδ and αD (n = 64) (Fig 5I–L). The results suggest that the co-expression of αDβ1 and Hbδ is a common feature of colorectal carcinomas. Normal colorectal tissue did not stain with Hbδ antibodies (Fig S6A), whereas weak staining was

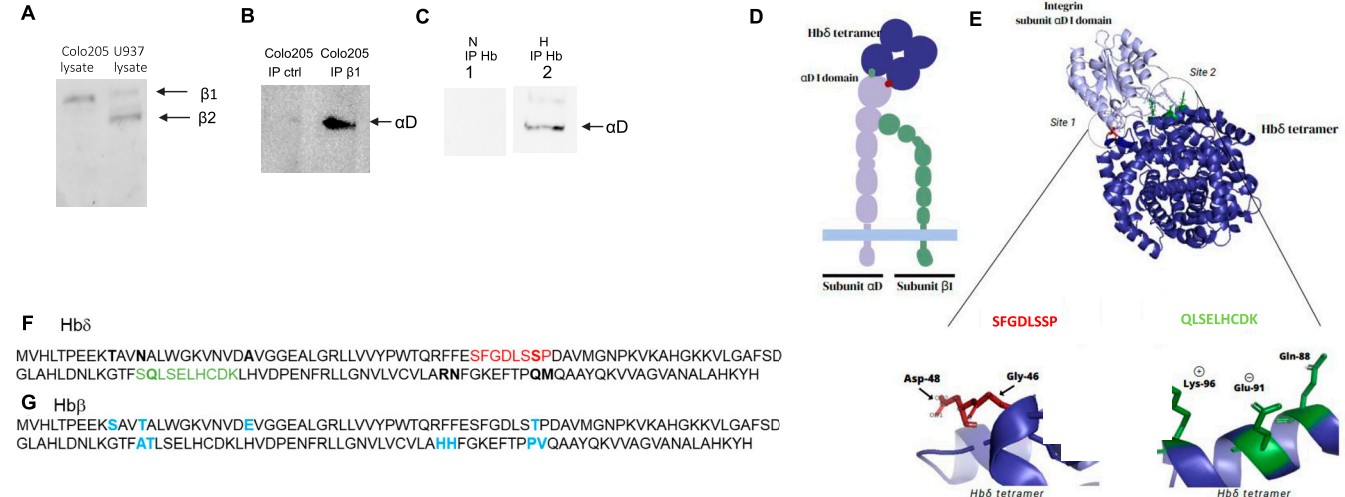

**Figure 3. Blotting and immunoprecipitation of αDβ1 and Hbδ-αDβ1 and prediction of the structure of the Hbδ-tetramer/integrin I-domain complex.**
**(A)** Blotting of β1 and β2 integrin polypeptides from Colo205 and U937 cell lysates. Colo205 cells contained only β1, whereas U937 cells contained both β1 and β2 integrin chains. **(B)** Blotting of integrin αD from a slab gel of immune precipitates obtained with control and integrin β1 antibodies (Santa Cruz and BD Biosciences) from Colo205 cells. **(C)** Immune precipitation with Hb antibody (Thermo Fisher Scientific) from cells grown under normoxia (N) and hypoxia (H), and visualization of integrin αD by blotting the gels with αD antibody. The integrin αD band was obtained from cells grown under hypoxia, but not from cells grown under normoxia. **(D)** Schematic structure of the αDβ1 integrin bound to the Hbδ tetramer (upper left), and predicted structure of the Hbδ tetramer bound to the integrin αD I-domain (upper right). The possible interacting peptide loops of Hbδ are below. **(D, G)** The sequence in site 1 is somewhat like the RGD sequence containing the amino acids (D, G) implicated in binding. The other important sequence is quite different. **(E)** Predicted structure of the Hbδ tetramer bound to the αD I-domain. The possible interacting peptide loops of Hbδ are below. **(D, G)** The sequence in site 1 is somewhat like the RGD sequence containing the amino acids (D, G) implicated in binding. The other important sequence is quite different. **(F)** Sequence of Hbδ used for cell transfections. The locations of the I-domain interacting sequences are highlighted in red and green, and the amino acids, which differ between Hbδ and Hbβ are highlighted in blue. **(G)** Sequence of Hbβ.
Source data are available for this figure.

obtained with integrin αD antibodies, primarily in interstitial macrophages (Fig S6B).

## Discussion

Colorectal cancer was the fourth most common cause of cancer death in the 1990s, but its incidence has since increased, being the most deadly cancer for men and second for women under the age of 50 (Ledford, 2024; Siegel et al, 2024).

Our results show that the Colo205 colorectal cancer cells grown under low oxygen conditions express both the previously unknown integrin αDβ1 and Hbs. The αDβ1 integrin forms a protein complex with Hbδ, which stimulates cancer cell proliferation via localized oxygen scavenging.

Recently, Zhang and co-workers reported that the avascular cartilage tissues of mice and humans express Hb (Zhang et al, 2023). Interestingly, Hb formed intracellular aggregates that resembled those that we observed. These findings indicate that low oxygen levels stimulate Hb production, resulting in functional Hb aggregates. In fact, turtles living in deep water under low oxygen express Hb polymers (Petersen et al, 2018).

Hbδ and Hbβ are very similar, differing in only 10 amino acids, and antibodies to the proteins often cross react. qRT-PCR and cDNA sequencing showed that Colo205 cells grown for 2 d under hypoxia express both Hbδ and Hbβ. By the use of recombinant Hbδ and cell transfection, we found that Hbδ binds to the αDβ1 integrin, affecting

cellular functions including adhesion and formation of integrin-Hb aggregates. Hbβ has been reported to have both growth supporting (Dean et al, 1983; Poddie et al, 2003; Zheng et al, 2017), and growth inhibitory properties (Maman et al, 2017; Luo et al, 2022), but a distinction between Hbβ and Hbδ was not performed in some of these studies. Hbδ seems more important here. The oxygen affinity of Hbδ is higher than that of Hbβ (Inagaki et al, 2000), which could explain why Hbδ is advantageous for malignant cells.

Genetic mechanisms for induction of HIF and many of the target genes have been reported (Maxwell et al, 1999; Ivan et al, 2001), but there has been a lack of understanding how proteins induced under low oxygen stimulate cell proliferation. Hypoxia can also upregulate other transcription factors, and, e.g., production of Hb in chondrocytes is dependent on KLF1 rather than the HIF1/2α pathway (Zhang et al, 2023). At least part of the Hb molecules at the cell surface is bound to integrin αDβ1. Importantly, oxygen uptake and cell proliferation depend on the co-expression of αDβ1 and Hbδ.

αDβ2 is a member of the leukocyte β2 integrin subfamily (Springer, 1990; Gahmberg, 1997). In leukocytes, it functionally resembles the αXβ2 integrin (Van der Vieren et al, 1995; Podolnikova et al, 2016; Aziz et al, 2017; Blythe et al, 2021). Integrins require both α- and β-chains for cell surface expression (Hibbs et al, 1990), and evidently, the lack of the β2 chain in Colo205 cells and colorectal tumors results in the formation of the novel αDβ1 heterodimer. There are other examples where an integrin polypeptide may combine with different integrin chains. The β1 polypeptide is known

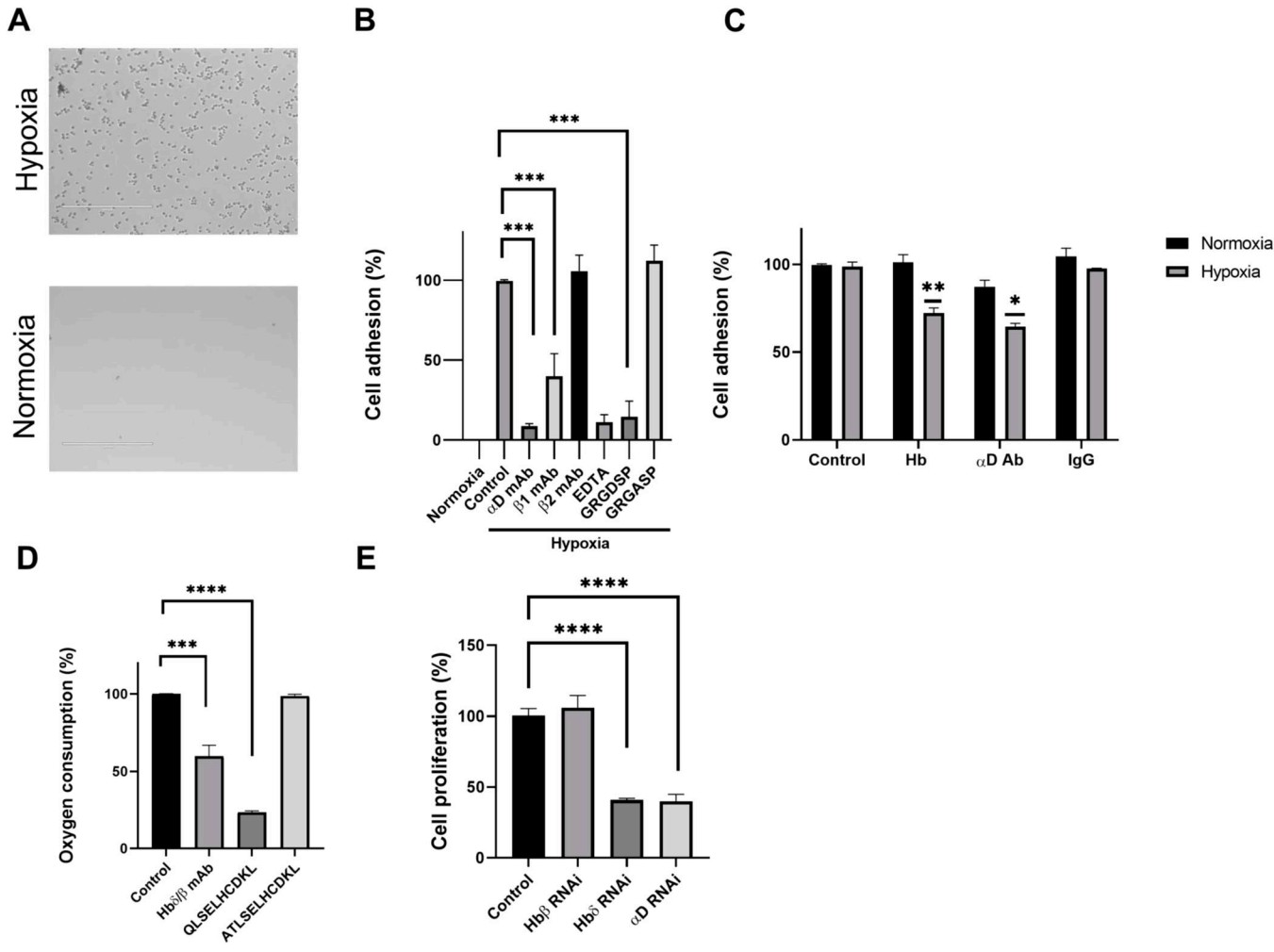

**Figure 4.  Adhesion, oxygen consumption, and proliferation of Colo205 cells.**
**(A)** Adhesion of cells to Hb grown under hypoxia and normoxia. Cells grown under hypoxia strongly adhered to red blood cell Hb, whereas those grown under normoxia did not. **(B)** The αD and β1 integrin antibodies effectively blocked adhesion, whereas the β2 integrin-blocking antibody showed no effect. The RGD-containing peptide also blocked cell adhesion, whereas the control peptide did not. **(C)** Binding of cells to fibronectin. Soluble Hb (250 μg/ml) and αD antibodies significantly blocked adhesion of cells grown in hypoxia. **(D)** Oxygen consumption by cells grown under hypoxia was inhibited by the Hbδ/β monoclonal antibody and the Hbδ-derived peptide QLSELHCDKL (400 μM), but not by the corresponding Hbβ-derived peptide ATLSELHCDKL. **(E)** Proliferation of hypoxia-grown cells was efficiently blocked by RNAi down-regulation of either Hbδ or integrin αD, but not by RNAi of Hbβ. ***$P < 0.001$, ****$P < 0.0001$, **$P < 0.01$, *$P < 0.05$.

to pair with numerous α-chains (Hynes, 2002), but the α-chains of the β2 integrin family members have not been known to do so. Our results now show that the αD polypeptide can combine both with the β2 and β1 chains. This means that human individuals can express at least 25 different integrins, instead of the previously known 24.

Peptides containing the RGD motif are efficient inhibitors of several integrins, including β1 integrins (Pierschbacher & Ruoslahti, 1984), but they have no effect on leukocyte β2 integrins, which bind to larger domains in intercellular adhesion molecules (ICAMs) (Springer, 1990; Li et al, 1993; Gahmberg, 1997). Our results showed that the fibronectin-derived peptide GRGDSP efficiently blocked αDβ1-dependent cell adhesion. The QLSELHCDKL peptide from Hbδ blocked oxygen consumption to a similar degree as the Hbδ/β antibody. Alanine substitution experiments using the AlphaFold2 program indicated the importance of Q87, and suggested that the

closely spaced and oppositely charged amino acids E90 and K95 are important for the interaction with the αD I-domain. The Q87 residue could play a role in mediating protein polymerization, as this residue, which corresponds to T87 in Hbβ, could explain the ability of Hbδ to compete in the polymerization of sickle-cell hemoglobin (Porcu et al, 2021; Zhu et al, 2022). On the other surface of the integrin-binding alpha helix, the invariant H92, also known as the proximal histidine, forms a covalent bond with the heme iron in all studied Hbs. This adjusts the distance of the iron atom to the planar heme group, and thus oxygen binding (Perutz, 1979). The ATL-SELHCDKL peptide from the corresponding Hbβ sequence was not active. The result supports the conclusion that Hbδ is primarily responsible for the effects of Hb.

Whereas this work was in progress, a large data analysis of cDNAs transfected into the colon cancer cell line HCT116 showed that the integrin polypeptide αD could be affinity-captured with

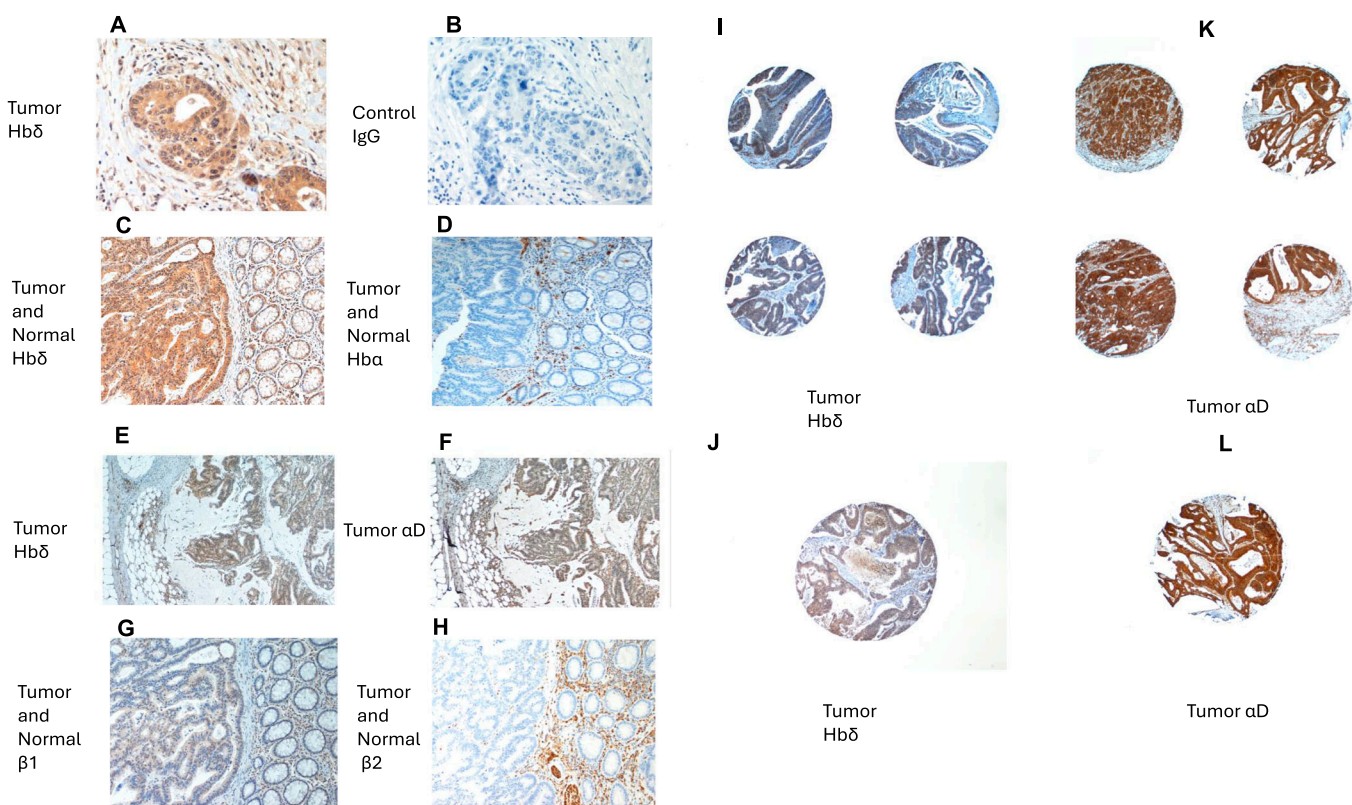

**Figure 5.  Expression and co-distribution of *α*D-integrin and Hb*δ* in colorectal cancers.**
**(A)** A cancer stained with Hb*δ* antibody. The tumor tissue is in the middle and lower right corner. **(B)** An adjacent section from the same tumor stained with mouse IgG. **(C)** Staining of a cancer with Hb*δ* antibody. **(D)** Staining of a consecutive section with Hb*α* antibody shows no staining of the cancer but positivity of red cells in the vessels. **(E)** Staining of a cancer with Hb*δ* antibody shows positive staining of the tumor. **(F)** Staining of a consecutive section of the same tumor with integrin *α*D antibody demonstrates co-expression of Hb*δ* and *α*D. **(G)** Staining with integrin *β*1 antibody. The cancer cells were strongly positive, as were the normal tissue to the right. **(H)** Staining of with an integrin *β*2 antibody. The tumor remained negative, whereas the leukocytes in normal tissue were strongly positive. **(I)** Four images of Hb*δ*-stained samples in a tissue microarray, magnification 100x. **(J)** Hb*δ* staining, magnification 100x. **(K)** Four examples of *α*D staining of a tissue microarray, magnification 100x. **(L)** Integrin *α*D staining. Original magnification 100x.

the Hb*δ* chain, suggesting that the proteins interact physically (Huttlin et al, 2021). The results did not identify any integrin *β* chain that would pair with *α*D, nor whether Hb*δ* is the only Hb that interacts with *α*D.

There appears to be a reason for the build-up of Hb*δ* multimers in hypoxia, which can change Hb allostery, and increase the oxygen affinity. Similar results have been reported with the *δ*4, Y4, and *β*4 tetramers, and other altered Hbs in thalassemias (Ager & Lehmann, 1958; Rochette et al, 1994; Sen et al, 2004; Fucharoen & Viprakasit, 2009; Chung et al, 2015; Hodroj et al, 2019; Shibayama, 2020). These facts may explain why malignant cells can take up oxygen under the hypoxic conditions, which are characteristic of tumors. Our results are supported by the recent public databases, COSMIC GRCh38 v.98 data (https://cancer.sanger.ac.uk/cosmic) and proteomics data (www.ebi.ac.uk/), which indicate overexpression of integrin *α*D and Hb*δ* in human cancer tissues and cell lines. An indication of the importance of integrin *α*D in colorectal cancer was obtained from the current National Cancer Institute CDC Data Portal. Of 437 cases of colorectal cancer, 110 showed integrin *α*D gene multiplication. The survival rate of these patients after 6 yr was about 30%, whereas that of the total patient group was 57%.

It is notable, that since the formation of the *Hbδ* gene in the eutherian lineage, its coevolution with the parental *Hbβ* gene has been complex (Moleirinho et al, 2015). Hb*δ* has become a pseudogene in some species, or a neglected member of the family, which is understandable if it augments cancerous cell growth.

Most important was our finding that down-regulation of *α*D*β*1 or Hb*δ*, or blocking the interaction between *α*D*β*1 and Hb with antibodies, efficiently inhibited cancer cell proliferation. The results show that both integrin *α*D and Hb*δ* are important in malignant growth, and the inhibition of tumor growth is at least partially due to the inhibition of oxygen uptake. It is certainly surprising that Hb is cell surface-located in cancer cells. The surface location of some of the Hb molecules bound to *α*D*β*1 in cells grown in hypoxia, could facilitate oxygen uptake into hypoxic cells and explain why the down-regulation of *α*D*β*1 and Hb affects cell proliferation. Our finding that soluble Hb inhibits Colo205 cell binding to fibronectin, indicates that the malignant cells could more easily disseminate from the primary tumors.

The finding that a high proportion of colorectal tumors express integrin *α*D and Hb*δ*, whereas normal cells show low or absent expression, has interesting clinical implications. Our preliminary

studies indicate that integrin αD and Hbδ are over-expressed also in breast cancer, lung cancer and melanoma.

Integrin αD and Hbδ/β antibodies could be useful in diagnostics of colorectal carcinoma. The presence of integrin αD or Hbδ fragments in fecal samples could indicate the presence of colorectal cancer. In addition, it is possible that integrin αD and Hbδ could be important targets for therapy. Knocking down the expressed molecules by CRISPR, RNAi, or by using specific integrin αD or Hbδ antibodies or peptide mimetics are obvious possibilities (Zeng et al, 2023). The CRISPR-Cas9 method has been approved for the treatment of β-thalassemia and sickle-cell anemia (Wong, 2023). Importantly, both integrin αD and HBδ are minor cellular components, and they may not be essential for life in adulthood. Lack or reduced levels of HbA2 in the absence of other mutations, does not necessarily cause clinical or hematological abnormalities (Steinberg & Rodgers, 2015). Therefore, off-target complications during treatment should not be a major concern.

## Materials and Methods

### Colorectal cancer cell lines and their maintenance in normoxia and hypoxia

The colorectal carcinoma cell lines (Colo205 and HT-29), and the monocyte/macrophage cell line U937, were obtained from ATTC, and stored in the cell depository of the Haartman Institute, University of Helsinki. The cell lines were authenticated by RNA sequencing, immunofluorescence and immunoblotting. The cells were cultured in RPMI-1640 medium under standard normoxic conditions at 37°C and 5% $CO_2$. In the hypoxia chamber, the oxygen level was set to 5% to mimic its physiological level in the circulation, and it was maintained with a compressed air dryer PNEUDRI DMOO2 (Domnick Hunter) in a Sanyo incubator (MCO-5M) at 37.5°C and 5% $CO_2$. Cell growth-dependent oxygen consumption was then induced in 75 or 25 cm$^2$ flasks (Sarstedt) flat surface down, and sealed with an adhesive polyester film (PCR Seal, 4titude) before adding the cap. The cellular oxygen level in cells under hypoxia was measured using 5 μM Image-iT Hypoxia Green Reagent (Thermo Fisher Scientific) and fluorescence emission at 520 nm. The cells were loaded for 30 min with the dye, washed and placed in growth medium and grown for 48 h in normoxia or hypoxia. Green fluorescence means that the oxygen level is below 5%. Cell numbers and viability were determined using an automatic cell counter (Countess; Invitrogen). The area of cells grown under normoxia and hypoxia were obtained by measuring 250 cells from five separate images using the Celleste 6.01 software (Thermo Fisher Scientific). Values are in arbitrary units.

### Blood and tumor tissue samples

We collected blood from healthy donors in heparin-containing tubes and kept the blood at 4°C on ice for at least 1 d before use in the oxygen consumption assays. Formalin-fixed and paraffin-embedded anonymous tissue specimens from patients with colorectal cancers were collected from the archives of HUSLAB

and the Department of Pathology, Haartman Institute, University of Helsinki, in accordance with the Finnish legislation and local ethical guidelines. Tissue archived sample examinations were performed with permission from the Finnish Medicines Agency Dnro FIMEA/2021/006901. Patients diagnosed with colorectal carcinoma underwent surgery at the Department of Surgery, Helsinki University Hospital. The mean patient age was 68 yr. All patients provided written informed consent upon inclusion in the study. The patient information, samples, and data, were handled and stored in accordance with the Declaration of Helsinki, and local regulations. The Surgical Ethics Committee of Helsinki approved the study protocol (permit 226/E6/06, extension TKM02).

### Synthetic peptides

The Hbδ and Hbβ-derived peptides were purchased from Tag Copenhagen A/S. SFGDLSSP and QLSELHCDKL were from the two predicted interaction sites with the integrin I-domain and were most affected by alanine substitutions according to AlphaFold2. ATLSELHCDKL is a corresponding peptide from Hbβ. VKHLQLDLES is a scrambled peptide with no activity. GRGDSP, GRGASP, and GRADSP were purchased from Sigma-Aldrich. The peptides were dissolved at a concentration of 50 mg/ml in $H_2O$.

### RNA sequencing and qRT-PCR

Total RNA (24 μg) was isolated from $5 \times 10^6$ Colo205 cells grown under hypoxia using the SV Total RNA Isolation Kit (Promega), and two extractions were pooled for further mRNA enrichment using dT-Sepharose. Sequencing was performed at the Institute of Biotechnology, University of Helsinki, and the data were processed using Illumina RNA-Seq Alignment Version 2.0.0.

cDNA was prepared from 1 μg of RNA using the iScript cDNA synthesis kit from Origene and 18 S RNA as a control. The primers for Hbβ, Hbδ, and αD integrin were obtained from Origene. The forward primer TGAAACCCTGCTTATCTTAAACAA, and the reverse primer TTATGTCAGAAGAAAGTGTAAGCAACAG, for Hbδ were from the regulatory sequence. TTGGACCCAGAGGTTCTTTGA and TCACTAAAGGCACCGAGCACT primers for Hbβ were obtained from the coding sequence (Manchinu et al, 2020). CFX maestro was used for quantification of qRT-PCR data and measurement of CT values.

### Immunoprecipitations and Western blotting

Before cell lysis, cell monolayers were washed with warm complete medium to inactivate integrin-associated proteinases (Björklund & Koivunen, 2005), and then with ice-cold serum-free RPMI-1640 medium supplemented with 1 mM PMSF (Sigma-Aldrich) and 1 mM benzamidine (Sigma-Aldrich). Cells were lysed with either 75 mM octyl β-D-glucopyranoside (Calbiochem) in TBS (pH 8.0) for Hbs, or radioimmunoprecipitation assay buffer (Sigma-Aldrich) for integrins, supplemented with proteinase inhibitory tablets (Roche). After scraping and a 30 min incubation on ice, lysates were centrifuged at 4°C for 30 min at 16,000g, to obtain the supernatants. Protein concentration was determined using the Bio-Rad Protein assay, and 30 μg protein was loaded per lane in either reducing (heated samples) or non-reducing conditions (non-heated

samples, for Hb), along with protein standards, and run at 80 V on a Mini-PROTEAN stain-free 4–20% precast gel (Bio-Rad). Immune precipitation of Hb was performed by incubating for 45 min at RT and overnight for integrins at 4°C, with antibodies bound to protein G Sepharose (Invitrogen). The immune precipitates were pelleted by centrifugation at 3,000$g$ for 3 min, and washed twice. Gel images were obtained using the Trihalo stain-free protocol of the Chem-iDoc Imaging system (Bio-Rad), and proteins were transferred to nitrocellulose membranes at 250 mA for 30 min (Hb), or 90 min (other proteins). Membranes were blocked with 5% milk, and incubated overnight at 4°C with primary antibodies. The membranes were then washed x 3 with TBS containing 0.1% Tween-20 and incubated for at least 2 h under gentle rocking at RT with the appropriate secondary antibody diluted 1:3,000. Finally, the membranes were washed, and blot images for IRDye-conjugated secondary antibodies were obtained using the Bio-Rad ChemiDoc Imaging System, after which enhanced chemiluminescence substrates (Thermo Fisher Scientific) were added to visualize horseradish peroxidase-conjugated antibodies. We used validated primary antibodies against all studied targets. For integrin αD, they were mAb (OriGene), rabbit pAb (Santa Cruz), rabbit pAb (Abbexa), or rabbit pAb against the αD cytoplasmic tail peptide CLEDKPED-TATFSGDDFSCVAPNVPLS (GenicBio Ltd). The integrin β1 chain was detected with mAb (Santa Cruz) or (BD Biosciences), and integrin β2 with mAb R2E7 for blotting, and 7E4 for inhibition of cell adhesion (Nortamo et al, 1988). Hb was detected with Hbδ mAb (Rockland), goat pAb (Thermo Fisher Scientific), Hbδ/β mAb (Santa Cruz), Hbα mAb (Santa Cruz, or Sigma-Aldrich), or Hbβ mAb (Santa Cruz). Secondary antibodies were anti-goat IgG (Pierce), anti-rabbit IgG (Cell Signaling), anti-mouse IgG (Cell Signaling), IRDye 680RD anti-rabbit IgG (Licor), and IRDye 800CW anti-mouse IgG (Licor).

### Determination of oxygen consumption

Cellular oxygen consumption was determined using a modified ex vivo assay in which cells were grown in the presence of human blood (Karjalainen et al, 2014). Colo205 cells were detached by scraping and suspended in RPMI-1640 medium at $3 \times 10^6$/ml with or without antibodies or peptide inhibitors. Hb and integrin antibodies were used at a concentration of 20 μg/ml. Whole blood was added to a final concentration of 10% into a 400 μl total volume of cells in 1.5 ml Eppendorf tubes. Samples (0.1 ml) were added in triplicate to 96-well plates (Greiner Bio-One), followed by the addition of 100 μl of heavy mineral oil (Sigma-Aldrich), an optically clear adhesive seal (Thermo Fisher Scientific) and a microplate lid. The plates were incubated in the hypoxia chamber, and the optical density was determined at 600 nm ($OD_{600}$). DeoxyHb formation was determined after overnight culture. The background $OD_{600}$ of the blood incubated alone was subtracted from the values. The reversibility of the $OD_{600}$ values was checked by incubation in normoxic conditions at the end of the experiments. To demonstrate the cell growth-induced formation of an $O_2$ gradient in cell culture flasks, cells were incubated with Image-iT Green Hypoxia Reagent (Thermo Fisher Scientific), for 60 min, after which the dye was removed, new complete medium added, and the flasks were sealed with polyester film and placed in the hypoxia chamber. After incubation, green fluorescence images were taken close to the tap of the flask, in the

middle areas, and at the other end, and quantitated using an Evos FL fluorescence microscope. Alternatively, the formation of an $O_2$ gradient was observed by incubation cells in the presence of 10% whole blood.

### Hbδ-GFP, Hbβ-GFP, αD-mCherry, and mCherry-β1 integrin transfections

Integrin αD cDNA constructs with C-terminal mCherry were obtained from the Genome Biology Unit supported by HiLIFE and the Faculty of Medicine, University of Helsinki, and Biocenter Finland. Briefly, the pENTR223.1 entry clone from the human ORFeome collaboration library was ligated into the mCherry-C vector using a standard LR reaction protocol (Thermo Fisher Scientific). Dr. M. Vartiainen provided the mCherry-C vector. The integrin β1 construct was constructed in a similar manner, but with mCherry at the N-terminus. Hbβ and Hbδ constructs were made with the GFP at the C-terminals. The constructs were sequenced at the Institute of Biotechnology, University of Helsinki. The Neon electroporation system (Invitrogen) was used to transfect the plasmids into Colo205 cells. Briefly, Colo205 cells were trypsinized and counted. 700,000 cells/reaction were transfected with five μg of plasmid DNA in a Neon 100 μl electroporation tip according to the manufacturer's instructions.

### RNAi treatment of cells

Colo205 cells were detached by scraping with a 10-ml pipette, centrifuged, and suspended in PBS at $15 \times 10^6$/ml. siRNA oligonucleotides targeting αD (SASI), Hbδ (SASI), and αM (control) were obtained from Sigma-Aldrich. The sense sequence for Hbβ was UCG GUG CCU UUA GUG AUG GTT, and the antisense CCA UCA CUA AAG GCA CCG ATT (Oligo TagCopenhagen). Two μl or 0.2 nmol per tube was heated in 10 μl of RPMI-1640 medium at 60°C for 2 min before adding 8 μl of TurboFect (Thermo Fisher Scientific), and then mixed with 100 μl of cells ($1.5 \times 10^6$) in T-25 flasks. The flasks were incubated in an upright position for 15 min at 37°C, after which 1 ml of RPMI-1640 medium was added, and samples were collected for microplate oxygen consumption and cell proliferation assays. After a 1 h incubation of the flasks under serum-free conditions, flat side down, 4 ml of complete medium was added, and the incubations continued for 2–3 d with the sealed flasks in hypoxia and non-sealed flasks in normoxia. After washing with serum-free RPMI-1640 medium, the cells were lysed as described above, and 30–50 μg of protein was used for SDS-polyacrylamide gel electrophoresis followed by Western blotting for αD integrin and Hbδ.

### Cell adhesion

Cell adhesion was determined in sealed 96-well plates in normoxic or a hypoxic chambers. Wells were coated with 10 μg/ml human ferrous HbAo (Sigma-Aldrich), 10 μg/ml recombinant Hbδ (Origene), or 10 μg/ml fibronectin (Sigma-Aldrich) in PBS overnight at 4°C, after which they were blocked by incubation with BSA and washed. The cells were detached in fresh complete medium, centrifuged, and suspended in complete medium at a density of $10^6$ cells/ml. The cells were preincubated with antibodies, Hb or peptide

inhibitors for 30 min before being pipetted ($10^5$ cells/100 $\mu$l) in triplicate or quadruplicate into the microwells, which were then sealed before adding the lid. After incubation for 60 min in hypoxic conditions, the film seal was carefully opened, and unbound cells were removed by washing with PBS. Bound cells were examined under a microscope and measured by incubation with p-nitrophenyl phosphate (Sigma-Aldrich) in 50 mM sodium acetate buffer (pH 4.9) containing 1% Triton X-100 (Sigma-Aldrich) until the OD405 was within a measurable range (checked from individual wells). Alternatively, bound cells were quantified metabolically by adding 20 $\mu$l of 3-(4,5-dimethylthiazol-2-yl)-2,5-diphenyltetrazolium bromide (5 mg/ml) (Sigma-Aldrich) per well and the OD590 read after solubilization of the dye.

### Fluorescence microscopy

Cells were cultured on round cover slips (Thermo Fisher Scientific) in six-well plates in normoxia or under film (hypoxia) for 3 d, and fixed with 4% paraformaldehyde for 40 min at 23°C. For HIF1$\alpha$ staining, permeabilization was performed with cold methanol. After saturation with BSA, cover slips were incubated with primary antibodies (diluted 1:1,000) overnight at +4°C followed by secondary antibodies for 1 h. Co-localization was studied using ImageJ, with the primary-secondary antibody pairs Hb goat pAb (Thermo Fisher Scientific) and Alexa Fluor 488-conjugated anti-goat IgG (InVitrogen), $\alpha$D integrin mAb (OriGene) and Alexa Fluor 647-conjugated anti-mouse IgG (Abcam). For integrins, we used $\alpha$D pAb (GenicBio) and Alexa Fluor 488-conjugated anti-rabbit IgG (Invitrogen), $\beta$1 mAb (Santa Cruz), or $\beta$2 mAb 7E4. For HIF1$\alpha$, the antibodies were mAb (Santa Cruz) and Alexa Fluor 488-conjugated anti-mouse IgG (Invitrogen). A Leica TCS SP8 was used for confocal microscopy, and an Evos FL or Zeiss Axioplan 2 for standard fluorescence microscopy. The figure for the visualization of the three-dimensional structure of the Hb-integrin cap was created using the mp4 program.

### Immunohistochemistry of colorectal tumors

Four $\mu$m thick sections of colorectal tumors were deparaffinized in xylene and rehydrated. Antigen retrieval was performed by microwaving the sections in 10 mM citric acid monohydrate for 3 × 5 min at 650 W. Endogenous peroxidase activity was blocked by treatment with 0.5% $H_2O_2$. The slides were incubated overnight at 4°C with the primary antibody in PBS containing 0.5% normal human serum. The same procedure was used for negative controls, except that incubation overnight in PBS without primary antibody. The reaction was visualized with 3-amino-9-ethylcarbazole (Vector Laboratories). Sample arrays of a large number of colorectal cancer tumors were performed in a similar manner. To show that major leukocyte integrins, were largely absent in carcinoma tissues, we used the monoclonal antibodies reported to work in immunohistochemistry: integrin $\alpha$L mAb MEM-25 (Antibodies-online), integrin $\alpha$M mAb (Novus), integrin $\alpha$X mAb 3.9 (Santa Cruz), $\beta$2 mAb MEM-48 (Abcam). Antibodies against integrin $\alpha$D, integrin $\beta$1, and Hb showing a positive staining were used as described above.

### Cell proliferation studies

Cells were detached by scraping or with trypsin–EDTA, and mixed with antibodies to be tested. The Hb$\delta$ peptides, GRGDSP, and the control GRGASP peptide were dissolved in complete medium and assessed for their effect on proliferation in triplicate wells in film-sealed 96- well plates under hypoxia, and non-sealed 96-well plates in normoxia. After culturing for 24, 48, or 72 h, the cells were studied microscopically and quantified using the phosphatase assay, as described above.

### Protein modeling using AlfaFold2

The known structures of Hb$\alpha_2\delta_2$ and Hb$\alpha_2\beta_2$ tetramers and the predicted structure of the I-domain of the integrin $\alpha$D were selected for modeling using AlphaFold2 (Jumper et al, 2021; Akdel et al, 2022), and subsequent visualization using the PyMOL Molecular Graphics System, Version 1.3, Schrödinger, LLC. The individual protein structures, including Hb$\delta$ monomer, were retrieved from the PDB, and then used as inputs for subsequent modeling steps, and to construct the Hb$\delta$ tetramer. The predicted structure of the complex between integrin $\alpha$D and Hb$\delta$ tetramer from AlphaFold2 was loaded into PyMOL, and in silico alanine scanning mutagenesis was performed at key residues to detect a decrease in the binding of Hb$\delta$ to integrin $\alpha$D.

### Statistical analysis

Results are presented as mean ± SD, unless otherwise indicated. Statistical significance was determined by $t$ test (n = 3 or 4 in most experiments) and $P$-values < 0.05 as shown in most figures.

## Data Availability

All data including the Trihalo protein stains of gels, uncropped and unprocessed images of all immunoblots, and histochemical array stains of patient samples will be archived in a permanent repository (See supplementary information). The experimental data and the results that support the findings of this study will then be available in the repository. The Public Data Base access is (https://portal.gdc.cancer.gov/).

### Ethical approval

Tissue archived sample examinations were performed with permission from the Finnish Medicines Agency Dnro FIMEA/2021/006901. The Surgical Ethics Committee of Helsinki approved the study protocol (permit 226/E6/06, extension TKM02). The consent of the patients were obtained.

## Supplementary Information

# Acknowledgements

We thank the previous and current members of the laboratory for their collaboration and helpful discussions, Dr. Roger Laine for comments on the text, and Mikko Liljeström for help with the confocal microscopy. This work was supported by grants from the Magnus Ehrnrooth Foundation to S Madhavan, E Koivunen, M Grönholm, and CG Gahmberg. The Finnish Society of Science and Letters supported CG Gahmberg. Further support was from the Walter och Lisi Wahls Stiftelse (E Koivunen), the Academy of Finland (LC Andersson, C Haglund, and CG Gahmberg), the Sigrid Jusélius Foundation (LC Andersson, C Haglund, and CG Gahmberg), the Liv och Hälsa Foundation (LC Andersson, M Grönholm, C Haglund, and CG Gahmberg), the Wilhelm and Else Stockmann Foundation (CG Gahmberg), and the Finnish Medical Society (LC Andersson, C Haglund, and CG Gahmberg).

## Author Contributions

E Koivunen: conceptualization, data curation, formal analysis, supervision, funding acquisition, methodology, and writing—original draft, review, and editing.
S Madhavan: conceptualization, data curation, formal analysis, validation, and methodology.
L Bermudez Garrido: conceptualization, data curation, software, formal analysis, validation, and methodology.
M Grönholm: conceptualization, data curation, formal analysis, supervision, validation, methodology, and writing—original draft.
T Kaprio: resources, data curation, investigation, and methodology.
C Haglund: conceptualization, resources, data curation, funding acquisition, and methodology.
LC Andersson: conceptualization, resources, data curation, formal analysis, supervision, funding acquisition, methodology, and writing—original draft, review, and editing.
CG Gahmberg: conceptualization, resources, formal analysis, supervision, funding acquisition, investigation, project administration, and writing—original draft, review, and editing.

## Conflict of Interest Statement

The authors declare that they have no conflict of interest.

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
