## [Reviewer comments · Life Science Alliance]

Life Science Alliance

Hypoxia favors tumor growth in colorectal cancer in an integrin α D β 1/hemoglobin δ -dependent manner

Carl Gahmberg, Erkki Koivunen, Sudarshan Madhavan, Laura Bermudez-Garrido, Mikaela Grönholm, Tuomas Kaprio, Caj Haglund, and Leif Andersson

DOI: <https://doi.org/10.26508/lsa.202402925>

Corresponding author(s): Carl Gahmberg, University of Helsinki

Review Timeline:

Submission Date:	2024-07-03
Editorial Decision:	2024-09-20
Revision Received:	2024-11-19
Editorial Decision:	2024-11-20
Revision Received:	2024-11-25
Accepted:	2024-11-25

Transaction Report:

September 20, 2024

Re: Life Science Alliance manuscript #LSA-2024-02925-T

Prof. Carl G Gahmberg
Division of Biochemistry and Biotechnology
University of Helsinki
Helsinki, PO Box 56 (Viikinkaari 5) FIN-00014
Finland

Dear Dr. Gahmberg,

Thank you for submitting your manuscript entitled "Hypoxia induces colorectal cancer through an integrin $\alpha\text{D}\beta\text{1}$ /hemoglobin δ cell surface complex" to Life Science Alliance. The manuscript was assessed by expert reviewers, whose comments are appended to this letter. We invite you to submit a revised manuscript addressing the Reviewer comments.

Thank you for this interesting contribution to Life Science Alliance. We are looking forward to receiving your revised manuscript.

Sincerely,

B. MANUSCRIPT ORGANIZATION AND FORMATTING:

Reviewer #1 (Comments to the Authors (Required)):

In their manuscript, the authors studied the colorectal tumor cell line Colo205, grown under reduced oxygen levels, and found that low oxygen concentration triggers the expression of the $\alpha\text{D}\beta 1$ integrin, which interacts with hemoglobin δ on the cell surface. This interaction leads to increased oxygen uptake and cell proliferation compared to cells grown under normal oxygen levels. The authors also observed a high co-localization of $\alpha\text{D}\beta 1$ and hemoglobin δ in both in vitro and in vivo settings, which warrants further investigation as a potential therapeutic target.

While the presented data is intriguing, additional controls are necessary to strengthen the findings. Firstly, the authors should verify the activation of hypoxic conditions in Colo205 cells by utilizing HIF1a nuclear localization. In Figure 2, although immunofluorescence may be a suitable method to investigate antigen cellular localization, utilizing western blot for quantitative comparisons of protein expression would be preferable.

It is important to illustrate the transfection efficiency for different constructs, such as mCherry- αD and GFP-Hb δ , as well as the interference efficacy for RNAi. Additionally, the authors should provide representative figures of adherent cells to support the cell adhesion experiment. Furthermore, to confirm the involvement of both proteins in adhesion, the authors should conduct a cell adhesion test in cells with overexpression of αD and Hb δ , either alone or in combination

Reviewer #2 (Comments to the Authors (Required)):

In their manuscript, Koivunen and colleagues propose a role for a novel surface protein complex formed by alphaD/Beta1 integrin and hemoglobin delta in colorectal cancer. The authors found that hemoglobin delta is expressed on the cell surface of a colorectal cancer cell line in close contact with integrin alphaD. The authors suggest that this complex may be involved in cell oxygen consumption and proliferation. This result is intriguing and interesting, but some weaknesses and lack of clarity prevent us from being completely convinced.

Main points:

1. My first point concerns the title of the manuscript. The authors state that "hypoxia induces colorectal cancer". Nothing in the manuscript supports this claim. The authors did not address any mechanisms of induction of carcinogenesis. Perhaps they meant "hypoxia favors tumor growth in colorectal cancer in an alphaDbeta1/hemoglobin delta-dependent manner".
2. All of the authors' claims are dependent on the colo205 cell line. The authors state without showing that some other cell lines were used. I think it is important to show these results, especially for HT-29. Indeed, the authors' claim is surprising because the mRNA level for alphaD and hemoglobin delta in HT29 seems to be null (CCLE database Broad Institute), at least in normoxia. Here we need more evidence to be convinced that this is a universal mechanism.
3. A methodological point : how do you maintain the oxygen level without introducing N2 ? why seal the culture plate ? How do you evaluate the oxygen gradient between 1-5%, how do you measure it ? Do you preincubate your media before the hypoxic experiment ? if so, at what oxygen concentration ?
4. In Figure 1, the authors claim that "some" cells are much larger. This statement could not be based on a few cells shown from cell aggregates. We need a quantification of the cell surface here.
5. The sentence "The fact that cells grew well under hypoxia suggests that cell adhesion and oxygen-binding proteins may be involved" is an overstatement. Non-adherent cells could grow perfectly in hypoxia and it depends entirely on the level of oxygen and the adaptation of cell metabolism. Here, the link between integrin and hypoxia should be justified on more solid grounds.
6. Where did the data in Table 1 come from? RNAseq? what is the threshold chosen to declare a gene expressed or not? The RNAseq data analysis is not present in the M&M section.
7. The sentence "Hb plots showed high molecular weight bands (not shown)" is cryptic. Show us the data and explain what you have found.
8. The authors found Hb alpha and delta in Table 1, but not beta (why?), but examined beta and delta by qPCR. Why?
9. The figures in Figure 2 are too small and the meaning of the last figure in Figure 2B should be clarified and why there is no colocalization in this figure (what is the scale bar for this figure). The term membrane cap is not clear and the authors should explain why the expression of alpha D and Hbdelta decreases at 24h compared to 9h. Nuclear staining would have been appreciated.
10. What is the significance of the result shown in Figure 3A? Figure 3C is just about the level of expression in normoxia (no

protein, no interaction...) or are the authors suggesting that the interaction is not the same between the two proteins in hypoxia? 11. In Figure 4 it seems that the interaction is not specific for any type of hemoglobin, is this correct? Do the authors evaluate the comparative binding strength between different hemoglobins? It seems that the authors suggest that alphaD transfection increases adhesion to Hbdelta, but these data are shown on two different figures EVFig1A and B, these should be combined if this is the correct comparison, are there any relevant differences here?

12. Do the authors have any idea about the mechanism that allows cells to benefit from the oxygen bound to hemoglobin delta? Did they try to target endocytic recycling? What about the ability of hemoglobin delta to bind integrin when cells are in an extracellular matrix, is there competition between binding sites?

13. The sentence "Hbdelta and alphaD RNAi ... resulted in ... decreased alphaD expression" is ambiguous, does Hbdelta targeting also target alphaD? if so, how?

14. How do the authors explain the effect on proliferation? is it because the culture depends on adhesion to Hb? is it related to a direct effect on the cell cycle? To what extent could it be extrapolated to in vivo cell behavior?

15. Regarding Figure 5, the legend to the figure is missing. What are the clinical characteristics of the patients studied in 5A and C, as they seem to have relatively differentiated tumors? The images are too small to assess some claims, such as the attribution of Hbalph only to erythrocytes. It is clearly not obvious. The same is true for E and F where there is no normal epithelial tissue to claim that they do not express alphaD and Hbdelta (only adipose and mesenchymal tissue is seen). The lack of beta2 expression in the tumors is surprising because normally you have a lot of myeloid cells in these tumors and they express beta2 integrins. Could the authors comment on this?

16. The claim of co-expression on the clinical samples should be supported by a co-expression study on the same sample, otherwise it is not very convincing.

Minor points

1. The authors should add line numbers to better assess where changes should be made in the manuscript.

2. The table should be provided as a table and not just text, otherwise it is not easy to read.

3. There is an error in the annotation of sub-figures in Figure 3, there is no Figure 3G.

4. the concentration of hemoglobin delta is missing in the M&M cell adhesion section.

Reviewer #3 (Comments to the Authors (Required)):

The manuscript by Koivunen et al. reports that the colorectal tumor line Colo205, grown under reduced oxygen tension expresses the $\alpha D\beta 1$ integrin, which forms a cell surface complex with hemoglobin δ . This resulted in high local affinity for oxygen, which increased cell proliferation compared with cells grown under normal oxygen tension. Moreover, $\alpha D\beta 1$ and hemoglobin δ showed high co-localization in colorectal carcinomas. The manuscript is written in a mostly comprehensive manner, although relatively short, the experimental setups appear to be sound and the proposed mechanism is potentially of interest.

However, given that the authors claim to report a hypoxia-inducible phenomenon, it will be important to determine the role of the PHD-HIF-VHL signaling and the impact of the study is likely to increase by addressing the following questions experimentally.

Major:

In general, it will be important to clarify to which extent are the hypoxia-induced events dependent on HIF transcription factors. What is the impact of HIF1a and/or HIF2a knockdown or knockout? Even more so, since the authors discuss HIFs later on in the manuscript.

Why did the authors choose 5% oxygen for hypoxia experiments? At this concentration oxygen is not limiting for mitochondrial respiration. What would be the impact of an oxygen concentration of 1.5% or lower?

Fig1A and Fig 1B: This is very observational for now. The authors should quantify the cell morphology, which is nowadays feasible in an automated manner.

Fig1D: This should be verified at the protein level e.g. Western Blot.

Fig2: The authors report an upregulation of alphaD after 2 hours of hypoxia. What is the proposed mechanism for such a rapid hypoxia-induced phenomenon?

Expanded view Figure 2A: The authors show that treatment with an HB antibody or peptide reduces oxygen consumption. Given that cancer cells tend to show a high glycolytic rate, how about glycolysis in this setting. How does this type of treatment affect cell viability and proliferation, compared to the knockdown shown later on.

Minor:

The authors describe experiments in hypoxia in the text but neither in the text, nor in the figure legends the precise oxygen concentration for the experiments are indicated.

Sometimes the very short abbreviations like "aD" for Integrin alpha D make the manuscript a bit hard to read and the

abbreviations are not consistent throughout the manuscript.

Thank you for the excellent handling of our manuscript. We appreciate the comments of the reviewers. They have done an excellent job and spent a lot of time to make suggestions to improve the manuscript. We think that we have been able to deal with most comments and suggestions. We have done new experiments and rewritten some parts of the text. Several figures have been changed as requested. The manuscript has been written according to the style of the journal.

We have made some small changes in the manuscript, which have not directly been dealt with by the reviewers.

Introduction p. 3, lines 18-25 added. We have added headings for the different topics.

Figs 1-5 are now more complete as suggested by the reviewers. We now have six supplementary figures and 1 video. In Fig. 5 C, D, G and H we have changed the texts to Tumor and Normal, because both types of tissues are there.

Reviewer 1. Our main focus in the manuscript is to study changes in cancer cells when they adapt to low oxygen pressure. We have shown (see below) that we really achieved hypoxic conditions. We have tested whether HIF-1 α locates to the nucleus under hypoxic conditions. Under normoxia we stained for the protein in the cytoplasm by immunofluorescence. After transfer to hypoxia the cells partially showed a nuclear localization of the protein, but there was a concomitant enhanced DAPI staining making conclusions difficult. The enhanced DAPI staining was not seen with control mAb, nor in normoxia. Whether HIF-1 α is bound in the extrachromosomal DNA (ecDNA) is unclear. Hypoxia can upregulate also other transcription factors, and e.g. the production of hemoglobin in chondrocytes is dependent on KLF1 rather than the HIF1/2 α pathway (see Discussion).

In Fig 1 we have used western blotting for integrin α D from cells grown under normoxia and hypoxia. Integrin α D and Hb δ are exceptionally sensitive to proteolysis and we need to pretreat cells with proteinase inhibitors before lysis as described in the Material and Methods. In Fig. S2 we show blots of recombinant Hb δ , Hb β and Colo205 cell Hb. Both show large aggregates under non-reducing conditions. Under reducing conditions we see no band. Therefore, we used qPCR to quantitate the mRNAs for α D and Hb β and Hb δ . qPCR is quite reliable.

We have determined the transfection efficiency for the different constructs and they were around 50%, page 5, second paragraph. The down-regulation of α D was about 70%. The down-regulation of Hb δ could not be determined due to protein aggregation, p 6, fourth paragraph.

We show photographs of Hb bound cells from hypoxia and normoxic cultures (Fig. 4A). Few cells bind to hemoglobin when grown in normoxia. See also Fig. 1A. The cells grown under hypoxia flatten out and are bigger. See Results, page 4 lines 21-29.. In Fig. S3, we see that α D over-expression increases cell adhesion. In all cases where we use hypoxia-grown cells, we see increased adhesion. In addition we see a strong increase in adhesion when both α D and Hb δ are over-expressed in hypoxia grown cells as compared to cells grown in normoxia (Fig. 4B).

Reviewer 2. The reviewer has brought up several issues to which that we respond.

1. The reviewer is absolutely right in that the title was not good. In fact, we had left out the word 'growth'. Anyway, the suggested title is excellent, and we have changed the title accordingly.
2. We have actually also studied to some degree the HT-29 and LS174T colorectal cell lines. We include here a Fig. S4 where we show that the HT-29 cells express α D integrin chain. Normoxia grown cells may contain quite low amounts of the integrin and Hb δ , which could be the reason that they are not found in the CCLE database. This is a qualitative analysis, not quantitative. Actually, we have studied several cases of breast cancer, lung cancer and melanoma and they seem to show increased expressions of α D and Hb δ . The HT-29 text is mentioned on p. 6, lines 22-24. The Image iT reagent is described on p. 9-10, first paragraph of MM, lines 30- 40, p.10, lines 1-5.
3. The oxygen level in the hypoxia chamber was chosen to 5% to mimic its physiological level in the circulation, and it was obtained by purging N2 gas with a compressed air dryer, which removes oxygen from the air. We seal the flasks and plates using a folio, which restricts gas diffusion like a basement membrane, and oxygen is then consumed. By use of the Image-iT Green Reagent, we could show that the oxygen level was below 5% in the hypoxia cultures and actually made a gradient. An oxygen gradient was also seen in the characteristic spectrum of red cell hemoglobin, when 10% blood was used (original data in the files).
4. We have now compared the sizes of the normoxia and hypoxia grown cells (Fig.1B). The hypoxia cells are much larger. Text p. 4, lines 21-27. MM p. 10, lines 3-5.
5. The fact that the cells flatten out when growing in hypoxia indicated that adhesion proteins are involved. The low level of oxygen in hypoxia suggested to us that oxygen binding proteins could be involved to ascertain an adequate level of oxygen. Hemoglobin was an obvious candidate, but of course, other oxygen binding proteins could have been possible. In a Nature paper we showed a long time ago that leukemia cells express hemoglobin (ref. Andersson et al.) and therefore we had some experience in the field.
6. Table 1 was obtained in preliminary experiments already 6 years ago when we specifically looked for integrins and red cell proteins in tumor cell lines. It is not quantitative or covering everything in the field, but it gave an initial motivation to study these proteins. The big surprise was the expression of integrin α D, and no expression of β 2. Another interesting finding was Hb δ . The experiment is described on p. 10,lines 27-30.
7. As mentioned above, we now show the Hb blots in Fig. S2. The molecular weights of Colo205 Hb is high under non-reducing conditions, perhaps corresponding to the Hb aggregates we see by

microscopy. Recombinant Hb δ also makes multimers, but Hb β shows up as a monomer. It is unclear, why the reduced form of monomeric Hb δ is difficult to detect.

8. The reason may be that the computer program did not distinguish between Hb β and Hb δ because they are so similar. Therefore, we assayed for both.

9. We have made bigger pictures, added bars, and included a video of the last figure of 2B (S Video). After growth for 24 h, hypoxic cells make large Hb condensates (possibly by phase separation), which attach to the α D β 1 integrin.

10. There we wanted to show that the β 2 antibodies work. U937 cells are positive but not Colo205 cells. Normoxia- grown cells express relatively low amounts of α D, and therefore no β 1 integrin comes down. It could also be possible that β 1 and α D chains do not interact in normoxia, but this is probably not the case.

11. As well as we know Hb δ is certainly active here, but we cannot exclude other hemoglobins. All red cell hemoglobin preparations contain small amounts of Hb δ . We do not have other purified hemoglobin polypeptides. The most similar is Hb β , but the peptide from Hb δ binds to the integrin, whereas the corresponding from Hb β , does not. This fact supports our conclusion that Hb δ is the most important integrin binder. For comparison, we did transfections with Hb β , but its expression was very low in Colo205 cells. We did not want to combine the Suppl. figures because they were done at different times and therefore not directly comparable.

12. We have actually now tested the binding to fibronectin. This was a good suggestion. Hemoglobin competed for the binding, which indicates that this property may be important in vivo. We have made a new Fig. 4C. Text see page 6, lines 19-21 and p. 9, lines 11-13.

13. This we do not understand. Perhaps they need each other for efficient expression or the other is degraded. Anyway, we have removed this sentence.

14. Probably, proliferation is stimulated by adhesion. This phenomenon has been described in the literature (anoikis). Increased uptake of oxygen should be stimulatory.

15. The figure legend was included, but we added the descriptions to the figures. We have studied more than 60 colorectal tumors and most are very similar with high α D/Hb δ co-expression. There is actually expression of integrin α D in normal colon tissue. We have included a new figure S6. The level as compared to the expression in tumors is low, and mainly due to interstitial macrophages. See p. 7, lines 8-10. Actually, one can see some β 2 expression in the tumors but the level is low as compared to the amount in blood vessels.

16. There is a problem if you want to stain with primary antibodies, especially monoclonals followed by anti-mouse antibodies. Therefore, it is a good method to use consecutive sections from the same tumors. We can easily see that the figures are from the same tumor in A and B, E and F and G and H. Leif Andersson has a 50- year experience of looking at tumor samples.

Minor points.

1. We have added line numbers.
2. This has been done.

3. Corrected. Actually, the E was lacking from the structure figure.
4. Has been added. 10 $\mu\text{g/ml}$.

Reviewer 3. The possible involvement of HIF1a and/or HIF2a is interesting and important, but not a major issue in this article. We have stained for the factors, but we do not see an obvious translocation to the nucleus. We used the hypoxia reagent, see rev. 2 point 3. The oxygen level is 1.5-5% in our hypoxia experiments. It is technically challenging to work with living cells at exact oxygen levels.

Fig. 1 A and B. This has been done. Page 4, lines 21-29. The western blots have been done. Fig. 1 C and S2.

Fig. 2. We do not know the mechanism, but translocation from the cytoplasm may be important for Hb δ . Integrin αD is present initially at low levels, but the synthesis is increased after transfer to hypoxia.

Concerning the effects of the Hb antibody. We have not studied glycolysis here. The cell viability is not much affected. The proliferation is decreased by αD antibodies (Fig S5) , but we have not studied the effect of an Hb antibody. It could be hard to find an Hb antibody that for example could inhibit the integrin-Hb interaction. The RNAi experiments show that both proteins are important for cell proliferation.

The precise oxygen concentration is hard to measure but in the hypoxia experiments it is below 5% as measured using the Image IT green reagent. See above.

We now use the nomenclature **integrin αD** as much as possible. It is sometimes problematic. An integrin consists of two polypeptides and αD for example is not an integrin (only half of one) whereas $\alpha\text{D}\beta\text{1}$ is an integrin.

We hope that we have been able to deal with most matters that the reviewers brought up. In addition to the effects on tumor cells by various reagents, we think that for integrin scientists, the description of the novel $\alpha\text{D}\beta\text{1}$ integrin is highly interesting, including its inhibition by RGD peptides.

The manuscript is written according to the LSA directions.

November 20, 2024

RE: Life Science Alliance Manuscript #LSA-2024-02925-TR

Prof. Carl G Gahmberg
University of Helsinki
Dept. of Biosciences
Division of Biochemistry and Biotechnology
University of Helsinki
Helsinki, PO Box 56 (Viikinkaari 5) FIN-00014
Finland

Dear Dr. Gahmberg,

Thank you for submitting your revised manuscript entitled "Hypoxia favors tumor growth in colorectal cancer in an integrin $\alpha\text{D}\beta\text{1}$ /hemoglobin δ -dependent manner". We would be happy to publish your paper in Life Science Alliance pending final revisions necessary to meet our formatting guidelines.

- please be sure that the authorship listing and order is correct
- please upload your main figures and your supplementary figures as single files
- please upload your table files as editable doc or excel files

Figure Check:

- please add weights next to all blots
- please add scale bars to Figure 2D
- please add scale bars to Figures 5, S1 and S6

LSA now encourages authors to provide a 30-60 second video where the study is briefly explained. We will use these videos on social media to promote the published paper and the presenting author (for examples, see <https://docs.google.com/document/d/1-UWCfbE4pGcDdcgzcmiuJI2XMBJnxKYeqRvLLrLS08s/edit?usp=sharing>). Corresponding or first-authors are welcome to submit the video. Please submit only one video per manuscript. The video can be emailed to contact@life-science-alliance.org

A. FINAL FILES:

B. MANUSCRIPT ORGANIZATION AND FORMATTING:

Sincerely,

November 25, 2024

RE: Life Science Alliance Manuscript #LSA-2024-02925-TRR

Prof. Carl G Gahmberg
University of Helsinki
Programme in Molecular and Integrative Biosciences
Faculty of Biological and Environmental Sciences
Helsinki, PO Box 56 (Viikinkaari 9C) FIN-00014
Finland

Dear Dr. Gahmberg,

Thank you for submitting your Research Article entitled "Hypoxia favors tumor growth in colorectal cancer in an integrin α D β 1/hemoglobin δ -dependent manner". It is a pleasure to let you know that your manuscript is now accepted for publication in Life Science Alliance. Congratulations on this interesting work.

DISTRIBUTION OF MATERIALS:

Again, congratulations on a very nice paper. I hope you found the review process to be constructive and are pleased with how the manuscript was handled editorially. We look forward to future exciting submissions from your lab.

Sincerely,
